# Seedling Responses to Organically-Derived Plant Growth Promoters: An Effects-Based Approach

**DOI:** 10.3390/plants10040660

**Published:** 2021-03-30

**Authors:** Simon Hodge, Charles N. Merfield, Wendy Y. Y. Liu, Heng W. Tan

**Affiliations:** 1The BHU Future Farming Centre, Lincoln 7640, Canterbury, New Zealand; charles@merfield.com; 2Faculty of Agricultural Sciences, Lincoln University, Lincoln 7647, Canterbury, New Zealand; wendy.liu@qiup.edu.my (W.Y.Y.L.); hwtan@stu.edu.cn (H.W.T.); 3School of Agriculture and Food Sciences, University College Dublin, Dublin 4, Ireland; 4School of Biological Sciences, Quest International University Perak, Ipoh 30250, Malaysia; 5Department of Cell Biology and Genetics, Shantou University Medical College, Shantou 515041, China

**Keywords:** algae, biofertilizers, biostimulants, New Zealand, organic farming, plant nutrition, seaweed extracts

## Abstract

Organically-derived biofertilizers and biostimulants, developed from harvested materials such as seaweed and waste from animal and fish processing, are currently the subject of much fundamental and applied research. These products have significant potential in reducing synthetic fertilizer inputs to horticultural, arable, and pasture-based agricultural systems, although there is frequently some ambiguity over the magnitude and consistency of any positive effects these products may have on plant performance. This study examined the effects of organically-derived plant growth promoters (PGPs) available in New Zealand on the early vegetative growth of 16 plant species maintained under glasshouse conditions. When applied as a root drench to low nutrient potting mix, the effects of the PGPs on seedling shoot growth were strongly related to the NPK contents of the applied solutions. Any positive effects on shoot growth were, on average, reduced when the seedlings were maintained in higher nutrient growing media. Applying the PGPs at concentrations twice, and four times, the recommended concentration, only caused further growth responses when the PGPs contained high levels of nutrients. Applying the PGPs as a foliar spray had negligible effects on shoot growth. Overall, the results of these trials suggest that the positive effects of applying some organically-derived PGPs on seedling growth are a function of the PGP nutrient content, and not due to any indirect effects related to phytohormone pathways or modification of rhizosphere microorganisms.

## 1. Introduction

A wide range of organic supplements, fertilizers, plant growth promoters [PGPs], and ‘biostimulants’ are now available to commercial horticulturalists and home gardeners. These products are frequently based on harvested organic materials, such as seaweeds, byproducts from fish processing and animal slaughterhouses, and the composting or fermentation of humic substances and compost ‘teas’ [1,2,3,4]. There are now numerous reports and reviews in the scientific literature which describe how such products increase plant growth, improve plant health, and enhance yield and quality of the final produce e.g., [1,5,6,7,8]. These PGPs have a general appeal to organic, sustainable, and regenerative growers on multiple levels: the biological origins of the product, the reduced reliance on synthetic fertilizers and pesticides, and, often, secondary claims of improving soil properties and microbial health [1,3]. Global sales of these products are, therefore, responding positively to a growing awareness of organic farming and the environmental harm caused by intensive farming, together with legislation restricting the use of inorganic fertilizers such as urea and ammonium nitrate. It has been estimated that the global market for seaweed-based fertilizers alone will exceed US $17 million by 2025 [9].

Although many scientific papers describe the enhancement of plant performance by organic PGPs, for some time it has been known that these effects can be highly variable, may not occur for all plants, and may only become manifest under certain environmental conditions [10,11,12,13,14]. Variation in plant responses to PGPs has, for example, been reported with respect to soil type [6], application method [15], storage conditions, and the duration of the study [16]. Additionally, some PGPs seem to be of benefit only when plants are experiencing adverse conditions, such as drought [17,18], salinity [19], or are challenged by pests and diseases [20,21,22]; but see [23].

Variation among studies examining the effects of the same products can also be considerable. For example, Edmeades [16] analyzed data from field trials assessing 28 organically-derived liquid fertilizers and found that some products resulted in both positive and negative effects on crop performance. It was suggested that the proportion of trials resulting in statistically significant positive effects was only in line with that expected by chance, and that many of the PGPs included in his data set did not actually contain sufficient plant nutrients or organic matter to improve plant growth when applied at the recommended concentrations. Earlier work involving New Zealand pasture plants also reported that liquid PGPs based on seaweed or fish waste had no consistent effects on pasture production, and that the magnitude of any observed positive effects could be predicted from the PGP nutrient concentrations [24]. A similar claim was made by McHugh [25] who suggested that many seaweed-based PGPs did not meet the nutrient requirements to directly increase plant growth.

Further ambiguity over the efficacy of organically-derived PGPs frequently arises because the exact mechanisms behind any effects remain unproven. Often the positive effects of PGPs are not claimed to be caused by direct provision of nutrients, but that the PGPs act as biostimulants and benefit plants by one or more indirect routes, such as improving nutrient-use efficiency, extending access to nutrients already present in the soil, improving soil ‘health’ and microflora, or by enhancing plant resistance to abiotic or biotic stresses [5,7,8]. In a similar vein, when PGPs do not increase primary plant performance measures, such as growth or yield, researchers often describe beneficial effects in terms of qualitative changes in plant chemistry or attributes of the harvest. For example, trials in Ireland examining commercial products derived from the brown seaweed *Ascophyllum nodosum* found no increases in yields of cabbages, onions and potatoes, but did find increased levels of health-promoting phenolic and flavonoid compounds [26,27]. Similarly, numerous studies investigating the effects of organically-derived PGPs on grape vines report positive responses in the chemical properties of berries and the extracted juice e.g., [28,29,30]. Although not always empirically demonstrated, it is frequently postulated that these modifications to plant chemistry are caused by plant hormone mimics or by actual phytohormones, such as auxins, cytokinins, and gibberellins, which influence multiple plant physiological pathways [16,31,32].

In a previous study, Hodge et al. [33] investigated the effects of several organically-derived plant foods and PGPs commercially-available in New Zealand on the growth of pinot noir cuttings. Using a low nutrient potting mix, the positive effects seen on the growth of the cuttings were strongly correlated with the nutrient content [NPK] of the applied PGP solutions. When the cuttings were grown in a high nutrient potting mix none of the products improved growth, suggesting that when plant nutrient requirements were already met additional nutrients could not induce additional plant growth.

The above example illustrates that when examining plant responses to PGPs it is important to examine effects under varying conditions before making general conclusions about product efficacy. Ideally, data should be gathered examining PGP effects on a range of plant species, under different basal nutrient regimes, and at a range of spatial scales. Subsequently, rather than basing conclusions regarding the effectiveness of these products solely on conventional statistical significance testing using raw data from individual trials, their efficacy is better expressed in terms of consistency, and repeatability of the magnitude and direction of effects obtained from multiple independent experiments e.g., [16,34,35].

In this study, to avoid the pitfalls associated with assessing PGPs under limited experimental conditions, we have adopted an approach whereby each PGP was tested on multiple plants, in growing media with different basal nutrient levels, and when using different application methods. As different plant species grow at different rates and have different foliage structures, summarizing the overall effects of PGPs based on raw growth data over many different plant species can be problematic. Therefore, we have estimated the response of the plants to PGP application by first calculating a standardized effect size (Cohen’s *d*) for each trial that not only considers the magnitude of any responses but also the variability of the data and the size of the trial in terms of replicate plants [36,37].

The primary aim of this investigation was to examine the efficacy of a range of PGPs that are readily available to New Zealand consumers. As a study system, we have tested the PGPs under glasshouse conditions for their effects on early seedling growth of typical horticultural or garden plants. To examine whether any observed effects were related to the nutrient content and composition of the growing substrate, PGPs were assessed when plants were grown in standard and low nutrient potting mix, and in topsoil obtained from an organic farm. To gain further insight into the workings of these products, additional trials were performed to examine the effect of application method (root drench and foliar application) and to examine whether plant growth responses were related to the concentration of the applied product. 

## 2. Materials and Methods

### 2.1. General Methods

Sixteen plant species were used in the trials: amaranth (*Amaranthus tricolor* L.), celery (*Apium graveolens* L.), chervil (*Anthriscus cerefolium* (L.) Hoffm.), coriander (*Coriandrum sativum* L.), bean (*Vicia faba* L.), lettuce (*Lactuca sativa* L.), lucerne (*Medicago sativa* L.), mizuna (*Brassica rapa* var. niposinica), onion (*Allium cepa* L.), pak choi (*Brassica rapa* var Chinensis), pea (*Pisum sativum* L.), radish (*Raphanus raphanistrum* subsp. *sativus* (L.) Domin), sage (*Salvia officinalis* L.), tomato (*Solanum lycopersicum* L.), white mustard (*Sinapis alba* L.), and wormwood (*Artmesia absinthium* L.). Seeds were obtained from King Seeds [Katikati, NZ] and Egmont Seeds [New Plymouth, NZ]. All the plants we used represent common and garden varieties of vegetables, herbs, green manures, or cover crops, although the details of actual cultivars or varieties were not recorded.

Trials were performed from July 2014 to February 2015 under glasshouse conditions (mean temperature: 20 ± 2 °C) at Lincoln University, Canterbury, New Zealand. Seeds were first sown in seed trays containing a low nutrient potting mix (80% compost and bark, 20% pumice, no fertilizer; Appendix B). When sufficient growth had occurred (e.g. 4-true leaf stage), the seedlings were transferred to plastic pots (8 × 8 cm square; 10 cm deep) containing the same low nutrient potting mix. One seedling was placed into the centre of each pot. All pots were placed on to individual plastic trays (10 cm diameter) so any excess liquid that drained through the potting mix could be re-absorbed.

All plants were lightly watered each day with untreated water using a hose with sprinkler attachment. No watering occurred on the day and following day that solutions of products were applied. At harvest, the above-ground parts of each plant were removed using scissors, placed into a paper bag, dried for three days at 65 °C and then weighed to obtain shoot dry weight. Due to the differences in the rates of growth of the different plant species, the times between sowing, application of products and harvesting differed among plant species (see Appendix A).

### 2.2. Plant Growth Promoting Products

Sixteen plant promoting products (PGPs) were tested. Although not always straightforward, we classified each PGP as either a general fertilizer, seaweed-based, or animal-derived (e.g., fish, blood, bone) using information on the packaging (Table 1). In all trials the growth of the plants treated with the PGPs was compared with the growth of plants treated with the same volume of water. We included some PGPs that are general inorganic plant fertilizers (e.g., Phostrogen N:P:K 14:10:27) to act as positive controls. Recommended dilutions of PGPs were obtained from information on the packaging (Table 1). Apart from one set of trials where the products were applied as a foliar spray (see below), the PGPs were always applied as a soil drench with 50 mL of solution being added to each pot.

Nitrogen concentrations of the applied solutions of each PGP were determined by using a Dumas style elemental analyzer (Elementar Vario-Max CN), where the sample is first combusted at 900 °C in an oxygen atmosphere. The combustion process converts any elemental carbon and nitrogen into CO_2_, N_2_ and NOx, and the NOx species are subsequently reduced to N_2_. These gases are passed through a thermal conductivity (TC) cell, in order to determine CO_2_ and N_2_ concentrations and the %C and %N, calculated from the sample weights. Other mineral nutrients were identified by first digesting the sample with nitric acid and then analyzing it on an Inductively Coupled Plasma Optical Emission Spectrophotometer (ICP-OES, Agilent, Santa Clara, CA, USA; Appendix C).

### 2.3. Root Drench Assays: Comparison of PGPs in Low Nutrient Growing Medium

Due to limitations of glasshouse space, and the quantity of seed available for some plant species, not all possible combinations of PGPs and plant species were tested. Nevertheless, using the low nutrient potting mix as a standard, each of the 16 PGPs was tested on a minimum of four plant species, and each of the sixteen plant species was assessed with at least four PGPs. Although the PGPs tested on each plant species were essentially selected at random, or at least in an *ad hoc* manner, the overall scheme was modified so that the same PGP was not always tested against plants in same taxonomic group (e.g., legumes, brassicas) or in the same ‘food’ class (e.g., herbs, vegetables, cover crops). The details of the combinations of PGPs and plant species, the replication used, and the time between treating of plants and harvesting (14–42 days), are summarized in Appendix A. In total, 148 PGP-plant combinations were used to compare plant growth in treated plants with that observed in water-treated controls. In each of these 148 combinations, between four and seven replicate seedlings were used for both the PGP-treated and water-treated control plants. The treatments assessed on the same occasion were arranged in a randomized complete block design, with respect to their distribution on glasshouse benches to help mediate any spatial biases in light or in temperature. When multiple PGPs were tested on the same plant species at the same time, the growth in the treated plants were all compared with the same set of control plants.

### 2.4. Root Drench Assays: The Effect of Growing Medium

Following on from the results of the main root drench trials described above, especially the lack of effects produced by some seaweed-based PGPs, it was decided to examine whether the effect of the PGPs was influenced by the growing media in which the seedlings were maintained. Therefore, we directly compared the effects of nine of the animal-based or seaweed-derived PGPs (Nourish, Phostrogen, Synerlogic, Baby Bio, Bounty, EcoFert, Seasol, Vertesea and Yates Seaweed) when using three different growing media. In addition to the low nutrient potting mix described above, trials were performed using a ‘high’ nutrient potting mix (80% compost and bark, 20% pumice, plus Osmocote^®^ slow-release fertilizer) and also using topsoil collected from an organic farm (Biological Husbandry Unit, Lincoln, NZ). The organic soil was first sieved (5 mm) and then heated to 100 °C for three days to kill weed seeds. This heat sterilisation of the organic soil would also likely impact its microbiological activity and its structure. However, these issues were considered secondary to our primary aim of providing a growing medium with naturally occurring nutrients and different mineral, clay and sand contents to those occurring in the potting mixes. Information regarding the pH and nutrient contents of all three growing media is provided in Appendix B. 

In these trials, five plant species were used (amaranth, celery, chervil, onion, pak choi), but, due to similar restrictions on glasshouse space and available seed as those described above, we did not assess all five plants, with all nine PGPs in all three-growing media. We selected combinations of PGP and plant species that we felt provided valuable and meaningful data with respect to different groups of PGPs, the efficacy of individual PGPs, and their interactions with the nutrients and form of the different growing media. In total, 92 combinations of plant, PGP and growing media were set up (each with four replicate seedlings) where plant growth obtained after addition of the PGP as a root drench was compared with that obtained in the control treatment in the same growing medium. Additionally, when a PGP was tested using the organic soil or the high nutrient potting mix, the PGP was also tested on the same plant species in the standard low nutrient potting mix at the same time. Further experimental details are given in Appendix A.

### 2.5. The Effect of Varying the Concentration of the PGP Root Drench on Seedling Growth 

In these trials only the animal-based and seaweed-based PGPs were examined to understand whether some of these products could produce growth responses not seen when applied at the recommended rate. Sixteen combinations of plants and PGPs were used to examine the effects of increasing the PGP dose on the growth response of seedlings. These plant/ product combinations were: Tomato (Bounty, Just Fish, EcoFert, Yates Fish); sage (EcoFert, Seasol, Synerlogic, Yates Seaweed); coriander (Actavize, Synerlogic, Vertesea, Yates Seaweed); lucerne (Actavize, Seasol, Vertesea, Yates Fish). 

The seedlings were grown in the low nutrient potting mix and five doses of each PGP were created by diluting the neat products to make solutions representing 0.25, 0.5, 1, 2 and 4 times the recommended rate. Water was used as a ‘zero’ rate control. There were four replicate seedlings of each dose for each combination of plants and PGP, producing a total of 384 seedlings. Further experimental details are given in Appendix A.

### 2.6. The Effect of Applying PGP Products as a Foliar Spray

The recommended dilutions of the 16 PGPs were tested for their effects on seedling growth when applied as a foliar spray on tomato, pak choi and mustard seedlings. The seedlings were grown in the low nutrient potting mix and sprayed to run off with a hand sprayer 30 days after sowing and then again at 40 days after sowing. Plants were harvested 50 days after sowing, 10 days after the second spray. For each plant species there were four replicates of each product and 12 replicates of the water control. 

### 2.7. Statistical Analysis

The summaries from each trial in terms of mean shoot dry weight, standard deviation and replicate numbers are given in Appendix A. All statistical analyses were carried out using Microsoft Excel and Genstat v21 (VSI International Ltd., Hemel Hempstead, UK). In general, the aims of this study were to examine whether the different PGPs, and different classes of products, produced consistent positive effects on the growth of the test plants. However, due to the range of plant species included in the trials, we found that we were faced with a range of different growth rates and different foliage structures. It was, therefore, desirable to standardize the effects of the PGPs in each trial, rather than use the raw shoot weight data. Therefore, the approach we have taken is to summarize the response of the seedlings to each PGP in each trial (plant × product combination) as a standardized effect size using Cohen’s *d*, which is calculated as [38],
$$d=\;\frac{{\bar{x}}_{PGP}-{\bar{x}}_{Water}}{S}$$
where $\bar{x}$ is the mean shoot dry matter and *S* is the pooled standard deviation, calculated as:$$S=\;\sqrt{\frac{\left(n_{PGP}-1\right)s_{PGP}^{2}+\left(n_{Water}-1\right)s_{Water}^{2}}{n_{PGP}+n_{Water}-2}}.$$

The variance of the estimated effect size *d* can then be approximated by,
$$V_{d}=\frac{n_{PGP}+n_{Water}}{n_{PGP}n_{Water}}+\frac{d^{2}}{2\left(n_{PGP}+n_{Water}\right)}$$
and the standard error of *d* calculated as the square root of *V_d_*,
$$SE_{d}=\;\sqrt{V_{d}}$$

Given that each PGP was tested on multiple plant species, an overall weighted mean effect size, along with 95% confidence, could also be calculated using a meta-analysis procedure based on residual maximum likelihood (REML, Genstat v21, see Appendix A). 

Subsequent REML mixed model procedures were performed using the estimates of Cohen’s *d* which were obtained from each trial as the response variable to assess the differences in the effects among the different classes of PGPs. For the trials performed in the low nutrient potting mix, we examined whether the effects of PGP root drenches were systematically related to the class of PGP class (fixed factor), with individual PGPs and plant species included in the model as random factors. For the trials comparing growing media on the plant responses to the PGP, the mixed models included PGP class and growing media as fixed factors along with the interaction term, and individual PGPs and plant species were included in the model as random factors. In the foliage spray trials, the PGP effects on foliage growth were tested for their relationship to PGP class and to plant species, with individual PGPs included in the model as a random factor. In the dose-response trials, Cohen’s *d* calculated for each combination of plant, PGP and dose was tested against a null hypothesis of zero effect using 95% confidence intervals.

The relationships between the concentrations of NPK in the 16 PGPs, and the relationships between these nutrient concentrations and the overall weighted mean effect of each PGP on shoot growth, were assessed using Spearman’s rank correlation.

## 3. Results

### 3.1. Root Drench Assays: Comparison of PGPs in Low Nutrient Growing Medium

The effects obtained for the 148 occasions when PGPs were applied as a root drench to seedlings grown in low nutrient potting mix are summarized in Figure 1 (see also Appendix A). Overall, the REML mixed-model analysis indicated that there were significant differences in the effects produced by the different classes of PGP (F_2,13_ = 5.53, *p* = 0.018), with the PGPs based on animal waste (Cohen’s *d* = 4.27) and the general PGPs (Cohen’s *d* = 2.46) producing greater average effects compared with the seaweed-based PGPs (Cohen’s *d* = 1.05).

When considering the general class of PGPs, only Biofeed, the organically certified ‘compost tea’, did not significantly increase plant growth (*p* = 0.125). All of the other general class of PGPs produced a significant positive effect on plant shoot growth (*p* < 0.001). Although, some individual trials using Nourish had a negligible effect on seedling growth (Figure 1a).

Although the PGPs based on animal waste produced the highest average overall effect, the results for individual products were highly variable (Figure 1a). For these PGPs, Yates Fish resulted in some trials where negative or negligible effects on plant growth were observed. Nevertheless, this product still produced an overall significant increase in plant growth (*p* = 0.006), possibly because one trial (using peas) produced a noticeably large effect (Cohen’s *d* = 8.7, Figure 1a). All the other PGPs in this class produced significant positive effects on plant growth (*p* < 0.001, Figure 1b).

From the seaweed-based PGPs, EcoFert (*p* = 0.796), Vertesea (*p* = 0.252) and Yates Seaweed (*p* = 0.222) did not significantly improve seedling growth overall, although they did have considerable effects (e.g., Cohen’s *d* > 2) on some plants in some trials. Although Seasol produced variable results, this PGP resulted in a positive overall effect on growth (*p* = 0.004). Baby Bio and Bounty resulted in positive effects in all trials which resulted in an overall significant effect on seedling growth (*p* < 0.001).

### 3.2. Root Drench Assays: The Effect of Growing Medium

The effects obtained when applying PGPs to plants maintained in different growing media are summarized in Figure 2. The growing media significantly influenced the response of the plants to the PGPs, in that the average effects seen when using the high nutrient potting mix were lower (Cohen’s *d* = 0.69) compared with those obtained when using the low nutrient potting mix (Cohen’s *d* = 2.52) and the organic soil (Cohen’s *d* = 2.8; F_2,76_ = 7.75, *p* < 0.001).

The reduction in the PGP effects in the higher nutrient growing media was largely a function of the results seen with the general and animal-based products. Similar to the results described above, most of the seaweed-based PGPs produced negligible or even negative effects on plant growth, regardless of the growing media, and, therefore, the interaction between growing media and class of PGP was also statistically significant (Figure 2; F_4,74_ = 2.78, *p* = 0.033).

### 3.3. The Effect of the PGP Concentration on Seedling Growth 

In the dose-response trials most of the animal-based PGPs produced significant effects on foliage dry matter accumulation at, or even below, the recommended dilutions of each product (Figure 3). The exception to this was Yates Fish, which produced no significant growth response even at four-times the recommended concentration (Figure 3).

Of the seaweed-based products, only Bounty produced a clear dose-related effect on seedling growth, with significant differences occurring at recommended rate and above. For the other seaweed-based PGPs the effects on growth were slightly variable, but when they were applied at four times the recommended dilution, EcoFert did improve growth of sage but not tomatoes, and Vertesea improved growth of coriander but not lucerne (Figure 3). 

### 3.4. The Relationships between Plant Growth and PGP Nutrient Content

Details of the full nutrient analysis of all the PGP applied solutions are presented in Appendix C. The concentrations of macronutrients NPK were all co-correlated (*r_s_* > 0.55; *p* < 0.03) and the applied solutions of nine products had relatively high levels of N (≥395 mg/L), P (≥119 mg/L) and K (≥77 mg/L), which, in order of N content (high to low), were Wonder-Gro, Baby Bio, Actavize, Phostrogen, Synerlogic, AlgoFlash, Just Fish, Bounty, Just Blood and Bone (Table 1). The Wonder-Gro solution had the highest N concentration (960 mg/mL) whereas the Phostrogen solution had by far the highest K concentration (1094 mg/mL) compared with the other solutions (≤388 mg/L, Table 1). As a group, the seaweed-based products tended to have relatively low nutrient levels except for Baby Bio and Bounty which are both fortified with NPK (Table 1).

Although the trends were not obviously linear, the weighted average effect (Cohen’s *d*) on plant growth of applying the PGPs as root drench to low nutrient potting mix was positively correlated to the concentrations of N (*r_s_* = 0.73; *p* < 0.001), P (*r_s_* = 0.80; *p* < 0.001) and K (*r_s_* = 0.74; *p* < 0.001) of the applied solutions (Figure 4).

### 3.5. The Effect of Applying PGP Products as a Foliar Spray

The effects on shoot growth of applying PGPs as a foliar spray were generally small and inconsistent (Figure 5) and, overall, there were no significant differences among the three classes of PGPs (F_2_,_13_ = 1.38, *p* = 0.286). The three plant species responded differently to application of the PGPs when compared with being sprayed with water (F_2,26_ = 6.28, *p* = 0.006), with the growth of tomatoes tending to be reduced when PGPs were applied and mustard responding most positively to the PGP sprays (Figure 5). Only foliar application of three PGPs, AlgoFlash, Baby Bio and Yates Fish, produced positive effects on foliage growth for all three plant species (Figure 5). 

## 4. Discussion

As with previous studies, the results of our trials confirm that the beneficial effects of PGPs on plant performance are highly variable, both when comparing individual products, classes of products and when testing the same product under different conditions [16,33]. In general, our results strongly suggest that increases in shoot growth obtained by applying PGPs as a root drench were related to the amount of macro nutrients (NPK) present in the applied solutions. For example, four of the seaweed-based products (EcoFert, Seasol, Vertesea, Yates Seaweed) had very low NPK concentrations and, on the whole, had very little effect on shoot growth. Similarly, Biofeed and Nourish in the general class of plant fertilizers, and Yates Fish in the animal-based group, also had relatively low nutrient content and also produced relatively small effects on plant growth. Conversely, some of the best performing, and most consistent products, were those with the highest NPK concentrations, such as Phostrogen, Wonder-Gro, Synerlogic, Actavize and Just Fish [33]. The results of the many combinations of PGPs and plants we tested strongly imply that the application of organically-derived PGPs does not, by default, increase the growth of plants, and that many of the PGPs we studied lacked sufficient levels of major nutrients to influence plant growth [16,24,25].

Further evidence of a nutrient-based mechanism was obtained from the trials performed using high nutrient potting mix and organic soil as growing media. When the PGPs were applied to growing media already high in basal nutrients, the magnitude, or even the manifestation, of positive effects on plant growth were reduced. This result suggests that when seedling nutrient requirements were already met by nutrients present in the growing media, then the addition of extra nutrients in the PGP could not increase growth further. Similarly, there was little effect in increasing the concentration of PGPs to twice and four-times the recommended rate if these products only contained negligible quantities of nutrients.

In a previous study, Hodge et al. [33] postulated that some PGPs not readily producing effects when applied to plant roots may be more effective when applied as a foliar spray, as for some nutrients, or for some plant species, absorption may occur more readily through foliage than through roots [39,40,41]. However, in the current study we found no evidence that applying products as a foliar spray consistently enhanced shoot growth, especially for those seaweed-based products that also produced negligible effects when applied as a root drench.

Our results depicting a lack of effect arising from the application of many seaweed-based PGPs are clearly not consistent with those reported in numerous other studies. For example, Aldworth and van Staden [15] reported that seaweed extracts applied as a root drench or a foliar spray improved seedling root and shoot growth in both cabbages and marigolds. Similarly, Mattner et al. [6] found that application of kelp extracts improved multiple aspects of seedling establishment and growth of broccoli plants under glasshouse and field conditions. We acknowledge that, even though our study strived to assess the efficacy of PGPs under a range of conditions, and on a range of plant species, there are still limitations to our investigation. For example, trials were performed under benevolent glasshouse conditions where biotic and abiotic stresses on plants were minimized, and the time frame of each trial was restricted to a maximum of 60 days between sowing seeds and harvesting of shoots [42]. Additionally, we measured effects after only one application of the PGP whereas many of the product labels indicated multiple applications should occur, especially during the growing season or when plants were fruiting. We also recognize that we only measured plant responses to PGPs in terms of shoot dry matter accumulation, and did not assess any qualitative changes in foliage chemistry, changes in gene expression associated with defense priming, or any associated changes in rhizosphere microorganism diversity and functioning e.g., [43,44,45].

Nevertheless, the results of the multiple trials we performed corroborate the results described in other studies and, at least, add further evidence that some organically-derived biofertilizers or biostimulants do not produce consistent positive effects on plant growth or yield e.g., [26,27,33,46]. Additionally, both Abetz [10] and Edmeades [16] implied that publication bias might also be responsible for creating apparent inconsistencies among studies, in that researchers may be less inclined to submit, and editors less likely to accept, papers reporting ‘non-effects’ of novel PGPs or biostimulants. We are not suggesting that all such products are valueless and that their popularity might be solely derived from the attractiveness of their environmentally-friendly image, often supported by aggressive marketing [47]: clearly some growers and home gardeners have been using these products for many years and seemingly obtaining acceptable results. For commercial growers, however, consistent and repeatable results are often essential for the continued use of a product, and a lack of these properties may only be discovered after several years of use [42].

The descriptions of products as biostimulants, plant growth promoters, plant growth regulators, and/or plant protective compounds is still open to much variation and interpretation, and can depend on the organic source of the product, the processes used in its manufacture, and the proposed mode of action [48]. There is additional debate over whether the biostimulant or PGP consists of the commercial formulated product or its active ingredient, and the implications this uncertainty has on the legislation and testing required for product registration [48]. During the analysis of our data, we had an issue with the classification of some PGPs because these products combine organic material from different sources or have an organic base fortified with inorganic plant nutrients. For example, Baby Bio contains seaweed extracts but is fortified with, for example, ureic nitrogen and phosphorous pentoxide, whereas AlgoFlash is described as formulated using ‘natural minerals’ but is not certified organic. Indeed, although our investigation originally aimed to investigate the mechanisms of organically-derived PGPs, as far as we can ascertain only Biofeed and Vertesea are certified for use on organic farms by the New Zealand organic certifier BioGro. 

A further issue arising from the study of specific products is that, often due to manufacturers merging or the acquisition of small companies by their larger counterparts, these products are frequently removed, reformulated, or rebranded. For example, the seaweed-based product EcoFert produced by the company Tui and available in New Zealand at the time of the study is no longer available, although the same company now produces an ‘Organic Seaweed Plant Tonic’ (The TuiTeam pers.com., 2018). Similarly, Yates NZ now offers an organically-based product combining both fish waste and seaweed extracts with inorganic nutrients (Yates Thrive Natural Fish and Seaweed + Plant Food Concentrate). In many ways, our comparison of the classes of PGPs are ambiguous because the patterns observed are a result of our classification process and the products we have included in the study, which are only a tiny subset of all the products available. If we had omitted some products or selected additional examples in each product class, then overall trends would likely change. Additionally, for example, if we had, with some justification, classified Baby Bio and Bounty as general-purpose plant foods, rather than seaweed-based PGPs, then the overall effect of the seaweed group of products would have been lowered even further. 

We have presented the results of our trials as standardized effects, rather than present summary plant growth data obtained in the different treatments. The use of standardized effect sizes to examine plant responses to PGPs allowed us to visualize and combine seedling growth data from multiple plant species, with a range of different sizes, growth patterns, and variability, in a single metric [16,35]. The calculation of standardized effects also allows the data to be included in meta-analyses to examine the effects of PGPs over multiple studies. We would encourage researchers to present summary statistics (mean, standard deviation, replicate numbers) or allow access to raw data that would allow the calculation of standardized effects such as Cohen’s *d*, Hedges’ *g*, and Glass’s Δ. With advancements in software and mixed model meta-analysis and meta-regression methods, the influence of moderator or explanatory factors on these standardized effects can now be examined further [38].

Although, presenting effects rather than actual shoot growth data allowed us to focus on the responses of the plants to the PGPs, and summarize the results of multiple trials within a single graphic, this resulted in the components of the effect, or lack of effect, becoming concealed. For example, the reduction of effect sizes in high nutrient potting mix often occurred because the control plants were considerably larger than similar control plants grown in low nutrient potting mix. We also found that some calculated effects were smaller than expected because the raw data were highly variable, not because of a lack of difference in mean growth between the control and PGP-treated plants. Nevertheless, these minor issues with using an effects-based approach can be overcome if raw data or summary statistics are also made available to readers as appendices or Appendix A.

## 5. Conclusions

In our trials, the primary growth response to the PGPs, measured in terms of seedling shoot growth, was strongly correlated to the concentration of macronutrients in the applied solutions, and many products lacked sufficient nutrients to affect plant growth. There is an increasing number of PGPs available to commercial growers and home gardeners, many based on organically-derived materials such as animal and fish waste and from seaweed or seaweed extracts. The results of this investigation underline the need for independent testing of these products on a range of plant species and over a range of different growing conditions and experimental scales, before meaningful statements regarding their effects on plant performance can be made.

## Figures and Tables

**Figure 1 plants-10-00660-f001:**
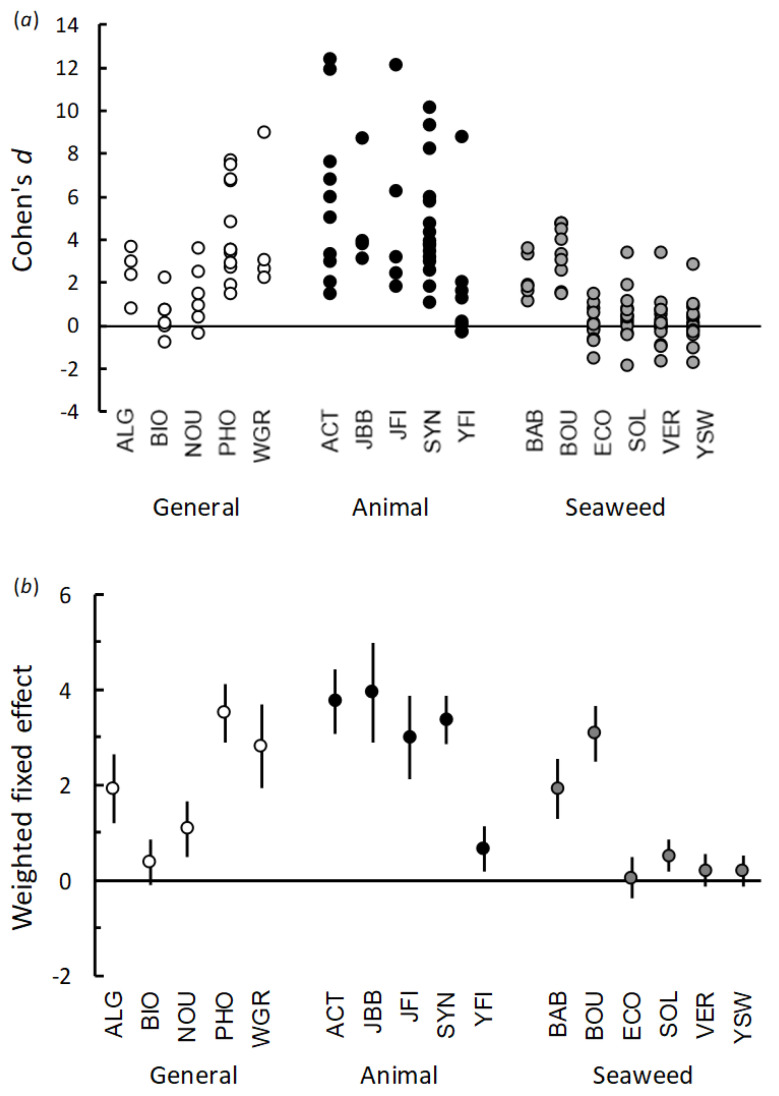
Effect (Cohen’s *d*) of applying different classes of PGPs (general, animal-derived, seaweed-based) as a root drench on plant shoot growth when plants were grown in a low nutrient potting mix. (**a**) Cohen’s *d* obtained in 148 trials; (**b**) weighted mean effects (± 95% CI) for each product estimated using REML analysis. Positive effects represent situations where the PGP caused an increase in shoot growth compared to plants treated only with water. See Appendix A for results of each individual trial. For product codes see Table 1.

**Figure 2 plants-10-00660-f002:**
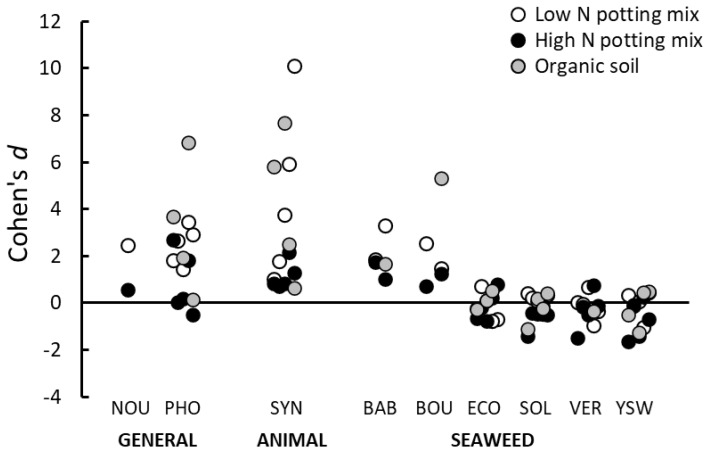
Effect (Cohen’s *d*) of applying different classes of PGPs (general, animal-derived, seaweed-based) as a root drench on plant shoot growth when plants were grown in a low nutrient potting mix, high nutrient potting mix or organic topsoil. Positive effects represent situations where the PGP caused an increase in shoot growth compared to plants treated only with water. See Appendix A for results of each individual trial. For product codes see Table 1.

**Figure 3 plants-10-00660-f003:**
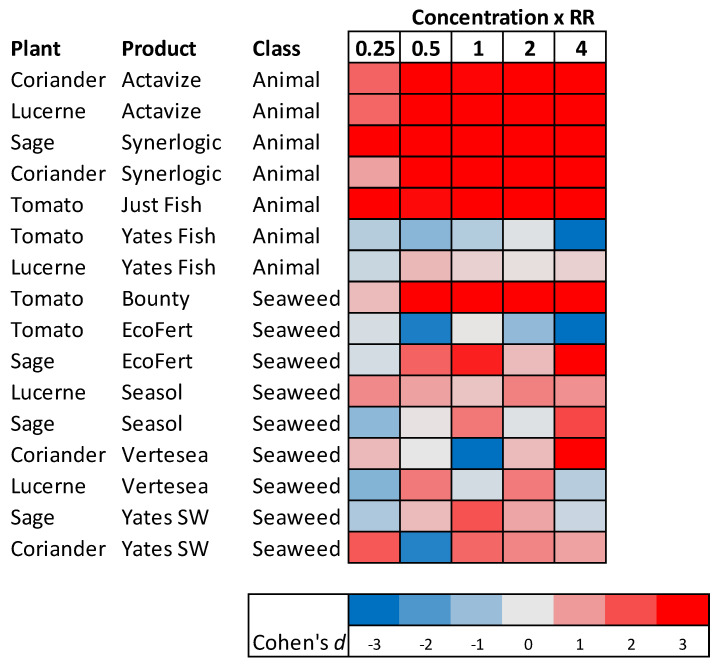
Heat map illustrating the effects (Cohen’s *d*) on plant shoot growth of applying seaweed-based and animal-derived PGPs as a root drench at different concentrations (x RR; recommended rate). Positive effects represent situations where the PGP caused an increase in shoot growth compared to plants treated only with water. See Appendix A for results of each individual trial.

**Figure 4 plants-10-00660-f004:**
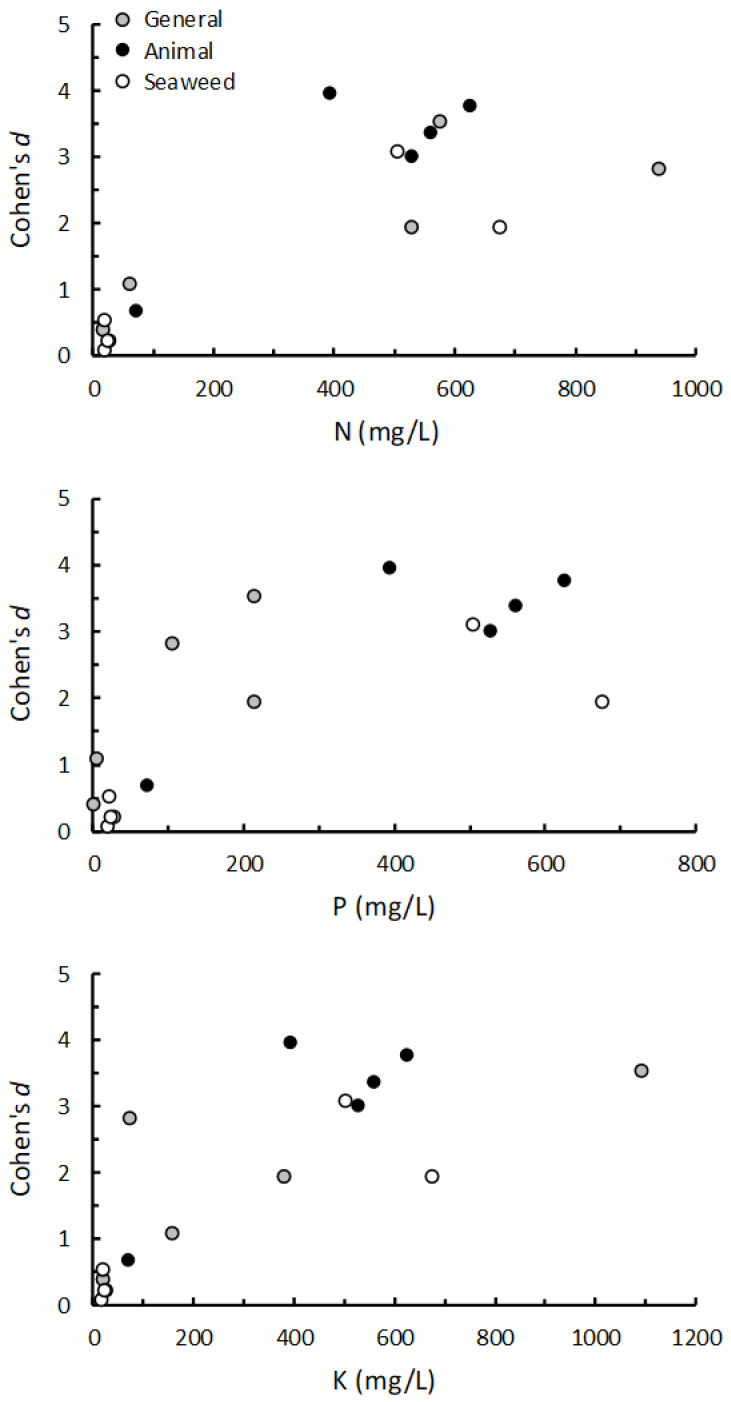
The relationship between the effect (Cohen’s *d*) on plant shoot growth of applying different PGPs (general, animal-derived and seaweed-based) and the NPK content of the applied solutions of PGPs. PGPs were applied as a root drench to low nutrient potting mix. The effect is the weighted mean effect of each PGP tested on multiple plants calculated using REML analysis (see Figure 1b; Appendix A).

**Figure 5 plants-10-00660-f005:**
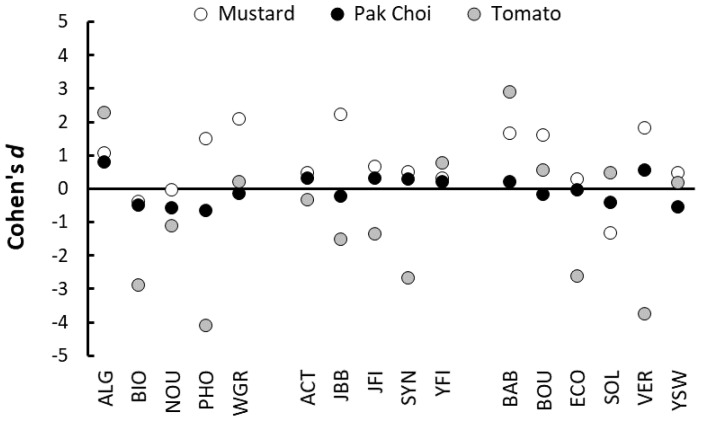
Effects (Cohen’s *d*) of applying different PGPs as a foliar spray at recommended dilution on the foliage growth of mustard, pak choi and tomato seedlings. The estimates of the effects were based on four replicate seedlings per plant species per product, and compared with growth observed when four control seedlings were sprayed with water. For the results of each trial, see Appendix A.

**Table 1 plants-10-00660-t001:** Summary of products tested in these trials, including class of product and dilution used (based on information provided on the manufacturers’ labels). NPK concentrations are those of the applied solutions. * g/L.

Code	Product	Manufacturer/ Supplier	Product Class	Applied Dilution (ml/L)	N (mg/L)	P (mg/L)	K (mg/L)
ALG	Algoflash	COMPO GmbH	General	6	531.5	215.3	382.1
BIO	Biofeed	Biofeed	General (organic ‘tea’)	20	19.2	1.3	20.8
NOU	Nourish	Kiwicare	General	10	62.4	6.8	159.7
PHO	Phostrogen	Bayer	General	5 *	577.7	214.9	1093.9
WGR	Wonder Gro	Watkins	General	2 *	940.4	105.8	76.7
ACT	Actavize	Fertiliser NZ Ltd	Animal (Fish)	6	628.1	231.3	387.5
JBB	Just Blood & Bone	The Warehouse	Animal (Blood/Bone)	5	394.9	118.6	265.5
JFI	Just Fish	The Warehouse	Animal (Fish)	5	529.4	146.2	310.6
SYN	Synerlogic	Fertiliser NZ Ltd	Animal (Blood/Bone)	6	562.0	187.8	355.3
YFI	Yates Fish	Yates	Animal (Fish)	5	72.7	39.2	11.6
BAB	BabyBio	Bayer	Seaweed	5	678.1	125.7	85.7
BOU	Bounty	Watkins	Seaweed	5	506.3	149.3	307.9
ECO	EcoFert	Tui	Seaweed	0.5 *	20.6	1.5	78.6
SOL	Seasol	Seasol Int.	Seaweed	10	22.2	0.1	144.9
VER	Vertesea	Fertilizer NZ Ltd	Seaweed	6	29.3	0.1	69.1
YSW	Yates Seaweed	Yates	Seaweed	10	26.1	0.4	38.3
	Water (control)	-	-	-	20.2	0.0	1.1

## Data Availability

Data from these trials can be found Appendix A.

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
