# Peer review of "Seedling Responses to Organically-Derived Plant Growth Promoters: An Effects-Based Approach"

_plants, 2021, doi:10.3390/plants10040660_

Round 1
Reviewer 1 Report
Thank you for submitting this article. I found it interesting and in the main the manuscript was well laid out and explained. I think the application of multiple PGPs against multiple plant species was interesting and the focus on biomass, while potentially narrow in its view does produce a manuscript that is actually useful from the perspective of agronomists and professional growers. There may be valid critique of not bringing the crops to a later development stage and possibly not evaluating plant quality criteria or even disease incidence, but on balance I believe the manuscript does add to the information on the impact of biostimulants particularly in the professional grower area and it is information that is, in my experience is required. The statistical approaches seem well laid out and comparing the means of the PGP effect against the negative control is valid, although in some occasions the SD is very high. I have included some comments and suggestions for edits below, however my main comment is that the explanation of how each PGP was assigned to a Plant species in experiment 1 needs to be better explained, was it random, were a number of different type of PGPs assigned to each plant species etc
Line 15: the word ‘is’ is missing after there or frequently
Line 47: Possibly worthy of reviewing and mentioning Yakhin et al, when discussing the perceived potential of the Stimulant market. It may also offer some guidance in the classification of products later in the document (Line 447). Yakhin et al., Bio stimulants in Plant Science, A Global perspective. Frontiers in Plant science (2017)
Line 147: 3 Days drying? No reference for approach taken and why 3 days? 48 hrs would be more typical?
Line 162-170: A general comment, in future studies an indication of the different types of N in the samples (NH4 v NO3) would be very informative, possible extractions with CaCl2/DTPA would give nice information on plant available N etc
Line 172-178: This section could be clearer. How were the plant species assigned to the 4 PGPs? Were they divided as a positive control, negative control etc. While I acknowledge the data is included in S1, how the decision of species v PGPs needs to be clearer.
Table 1: The assignment of Baby Bio and Bounty as seaweed based PGPs is not supported by their NPK content. I concede this is mentioned in the discussion, but it is not well explained. Its NPK is atypical for the group, and I would contend it should be grouped with the ‘General category’, however I am open to the authors rebuttal on this point.
Line 199: The sterilisation of the top soil from weed seeds would fundamentally alter the microbiological content of the soil and the structure would be affected. Has this method been used previously, particularly when using it a pot – A reference on this approach would be appropriate.
Line 204: A explanation of how plants were assigned PGPs and vice versa.
Line 273: Upon review of the supplementary data, reach interaction between a plant species and PGP uses a pooled water control, so I would not describe them as 148 trials, I would content that each plant species could be described as a trial with 8 treatments
Line 424: It may be appropriate to state if any of the instructions on the products tested indicate on their instructions of use if multiple applications are required
Line 458-461: Comments on product availability - Not sure this is relevant/ of interest to international readers (Delete)
Figure 3: I do not think ‘compound’ is the correct term here – product?
Reviewer 2 Report
What an absolute joy to read this manuscript. Not only was it well-written, it was clear, conscise, well-organized, and thorough. Every question I had as I read the manuscript was answered by the authors later on. I am in the rare position of having no suggestions for revision or improvement. If anything, I find the authors to almost be too self-critical about their approach and its possible limitations. Don't be! This is well-done research with high practical impact. I look forward to recommending it to others when it is published.
Reviewer 3 Report
The article is well conceived, conducted and written. It needs to be as the results presented do challenge some notions of plant "biostimulants".
I cannot comment specifically on the indices employed as I have not used them myself. However the data analyses seem entirely reasonable and so too conclusions drawn. The authors are thorough in presenting balanced discussion and caveats to the data set. The article is worthy of publication to create debate.
The authors provide good guidelines for the better understanding of plant stimulation products.
I have appended a file with some comments for the attention of the authors which I hope they will find constructive
I look forward to publication.

Reviewer 4 Report
This manuscript described the effects of organically-derived plant growth promoters on the vegetative growth of 16 plant species. The work that was done is quite impressive and manuscript is well written, I have only 2 comments:
- 1) For better orientation in the text, provide a complete list of abbreviations.
- 2) I think that the presentation of data using Fig.1, 2, is confusing and the reader is less oriented in the results. It is not clear what plants were used in the experiment and this information is difficult for the reader to find (in Appendix, Supplements). The model in Figure 3 (heat map illustration) seems better to me. It clearly illustrates the used plants, the compound and the PGP group. I recommend to present the data in Figures 1 and 2 using the similar Heat maps or extend figure legends.
Reviewer 5 Report
The whole paper is based on confusingly high number of different experiments without clear scientific plan in their conduction. It seems that lack of clear experimental structure, which is covered with good written English and visually attractive figures, doesn't deliver scientific novelty as well as scientific excellence which can be expected of papers published in Plant journal as WoS Q1 journal.
